# Plug and Play Multi-Organ Chips: Integrated µGaskets for the Facile and Reversible Connection of Individual Organ-on-Chip Modules

**DOI:** 10.3390/mi16111251

**Published:** 2025-10-31

**Authors:** Hannah Graf, Martin Gaier, Caroline Culp, Peter Loskill

**Affiliations:** 1NMI Natural and Medical Sciences Institute at the University of Tübingen, Markwiesenstrasse 55, 72770 Reutlingen, Germany; hannah.graf@nmi.de (H.G.); martin.gaier@nmi.de (M.G.); caroline.culp@nmi.de (C.C.); 2Department for Microphysiological Systems, Institute of Biomedical Engineering, Eberhard Karls University Tübingen, Waldhoernlestraße 22, 72072 Tübingen, Germany; 33R Center Tübingen for In Vitro Models and Alternatives to Animal Testing, Waldhoernlestraße 22, 72072 Tübingen, Germany

**Keywords:** microfabrication, microfluidics, organ-on-chip, multi-organ chip, modular connection, thermoplastic elastomer, burst pressure, chip connection, tight sealing

## Abstract

Multi-organ-chip (MOC) models provide a plethora of auspicious opportunities to replace current in vitro and in vivo models for a more systemic investigation of human (patho-)physiology for drug development and personalized medicine. Integration of individual organ tissues into a systemic circulation remains a major challenge for their implementation/application. Modular ‘mix-and-match’ connection strategies are beneficial in their flexibility for individual organ-on-chip (OoC) module designs, and their connection and experimental timelines, but yet lack a facile implementation/realization without the addition of external connectors and dead volume. We introduce a novel concept for the flexible plug and play integration of OoC modules to an MOC platform by integrated µGaskets. The thermoplastic elastomer (TPE)-based µGaskets provide a highly robust and simultaneously easy connection of customizable tissue models. We characterized the facile fabrication of connection chips equipped with µGaskets and proved their functionality and durability in different burst, pressure and reusability tests.

## 1. Introduction

Microfluidic technology has undergone rapid development in the last two decades. Predominantly used in analytical chemistry at first, manipulation of small volumes and the precise control of fluids became increasingly important for further fields of application [1,2]. Key developments in microsystem technology also formed the technical foundation for the emergence of microphysiological systems (MPSs), bridging the gap between cell culture and in vivo biology. MPSs, particularly organ-on-chips (OoCs), are assuming a growing importance in biomedical and pharmaceutical research and have already been successfully transferred into industrial applications, making a direct impact on patient care. Recent announcements from the FDA to include OoCs into their roadmap highlight their growing relevance and the trend to replace current in vitro and in vivo models for non-clinical testing [3].

Most OoCs emulate single organs or tissues. However, recreation of a single organ is often not sufficient to predict human (patho-)physiology or drug response as many processes depend on crosstalk across different organs [1,2,4,5,6].

Multi-organ chips (MOCs) offer a path to overcome these limitations by enabling the functional integration of multiple interconnected organ models. This allows for a more comprehensive investigation of human physiological and pathological processes, making MOCs particularly valuable for drug development and personalized medicine [7,8,9]. They enable systemic insights, for instance, in the synergistic organ interplay between the pancreas and liver in the regulation of glucose levels, which is already successfully replicated in different MOCs [9,10,11]. Furthermore, organ connection facilitates a more profound understanding of mechanistic interactions in disease progression and response to drugs and their absorption, distribution, metabolism and excretion (ADME) [6].

Despite the pressing demand of these multi-organ devices for the pharmaceutical industry and clinical research, the technical realization of organ model interconnection remains a key challenge for their development and implementation to date. A crucial consideration is the scaling of individual tissue units. The most common scaling approach is allometric scaling which uses ratios for organ size and body mass [7,8,9]. However, for understanding human PK/PD processes, especially in drug development, it is critical to account for the cells’ functionality and metabolism as well as the circulating volume in systems with organ–organ communication. Thus, attention has been drawn to methods measuring the metabolic activity of cells and designing appropriate residence times of the circulating fluid between the different tissues [8,10]. With respect to the actual physical connection, current methods can be categorized into monolithic/static and modular approaches [11]. In the monolithic approach, tissues are permanently inbuilt in a single microfluidic device and interconnected via a defined fluidic pathway. One of the first studies using a monolithic MOC was made by Shuler et al. for toxicity screening on a lung, liver and ‘other tissues’ device [12]. Such monolithic MOC devices eliminate the need for separate connectors, enabling single-platform fabrication and reduction in time and labor. Importantly, both the internal dead volume and the overall setup size are minimized [13,14,15,16]. However, modification of a single component involves an iteration of design, fabrication, and testing of the entire system. Furthermore, this strategy is biologically demanding, being restricted in terms of tissue-specific experimental timelines and protocols as prematuration and loading cannot be performed independently. Moreover, failure of a single component causes the malfunction of the entire system, reducing the overall success rate of this model concept [11].

As part of the monolithic approaches, semistatic concepts effectively bypass the confined possibilities for independent tissue preparation by the adoption of Transwells^®^ [11]. Individual integration of desired tissue wells expands the scope of application but is still limited in both the fluidic pathway and the complexity of the tissue chamber.

Given the number of challenges faced by monolithic (static and semistatic) approaches, modular “mix-and-match” strategies are gaining increased attention. Modular MOCs integrate individual OoC units while maintaining desired experimental flexibility and allow already well-established custom modules to be used [14]. Connection via tubing is a straight-forward approach when using existing tube adapters. However, considerable dead volumes and absorption of small molecules by tubing materials are major drawbacks that reduce applicability of these systems [17]. Small fluidic connectors such as barrel fittings or capillaries to stack individual MPS modules together offer a viable solution to increased dead volumes in the system [18,19]. However, connectors introduce external materials into the biological system and are thus a source of contamination and bubbles and can lead to pressure peaks during connection.

Modular connection by compression sealing utilizes soft intermediate layers to tighten the chip interface upon clamping pressure. The Tetris-like (TILE) platform of Ong et al., for instance, is tightened via a PDMS module pressed together using magnetic force [20]. Although still very frequently utilized, PDMS provides major disadvantages particularly for MOC studies where absorption of small molecules can confound experiments. While surface adsorption is a general problem of many materials including polymers, the partitioning of the hydrophobic molecule into the bulk material of PDMS due to its spongy nature is a major limitation. Particularly in combination with the unfavorable surface-to-volume ratio of microfluidics, this can lead to complete depletion of drug compounds or signaling molecules from the circulating media. In contrast, oil-free formulations of TPE were shown to have a markedly reduced molecule absorption, making it a suitable material for drug development studies [21,22,23,24]. Similarly, commercial O-rings and other approaches such as 3D-printed gaskets as tools for a leak-free interface of microfluidic chips are limited in biocompatibility and dead volume [25]. These technical constraints are the main hurdles limiting a broader adoption of modular concepts for MOCs.

Here, we address these technical challenges, introducing a novel thermoplastic elastomer (TPE)-based strategy for the flexible plug and play connection of OoC modules via directly integrated µGaskets, enabling modular connection with minimal dead volume. The strategy is specifically designed to connect customizable OoCs in a fast and user-friendly manner without the need for additional external connectors. It allows for (a) flexible and reversible interconnection of user-defined OoC modules, (b) a fast and easy fabrication process, and (c) a highly robust connection procedure and performance for a wide range of microfluidic pressures. We present and characterize an integrated microfabrication approach to produce connection chips equipped with µGaskets made from TPE styrene–ethylene/butylene–styrene (SEBS) and demonstrate the robustness of the connection mechanism. Burst tests revealed sustained pressure magnitudes beyond those used in classical OoC applications and the connection setup demonstrated user-independent functionality and reusability. Our novel connection strategy distinguishes itself through its minimalistic layout and easy adaptability to a wide range of different organ chips.

## 2. Materials and Methods

### 2.1. Fabrication of µGaskets

The connection chip and master mold were designed in the computer-aided design software AutoCAD (Version V.116.0.0, Autodesk GmbH, München, Germany) and SolidWorks (Solidworks 2019, Dassault Systèmes, Vélizy-Villacoublay, France). Polymers, including polymethyl methacrylate (PMMA) (PLEXIGLAS^®^ Film 99524, Gilching, Germany) and polyethylene terephthalate (PET) (3M™ Scotchpak™ 1022 Release Liner Fluoropolymer Coated Polyester Film, Bangalore, India) were used for fabrication. Commercial SEBS pellets (Mediprene OF400M, HEXPOL TPE AB, Åmål, Sweden) were extruded to a 750 µm thick foil by an external service provider (Fraunhofer IVV, Freising, Germany).

The connection chip comprises a 750 µm thick SEBS sheet which entails the µGaskets and in- and outlets on the top surface, and a 250 µm thick PMMA layer for the channel, featuring a width of 400 µm and height of 250 µm. The bottom layer is a 750 µm SEBS sheet bonded to a PET foil, allowing for optical accessibility and protection from dirt (see Figure 1).

The µGaskets are produced via hot embossing using a master molding tool consisting of two steel-based molds for double-sided molding designed in a CAD program (Solidworks 2019, Dassault Systèmes). The top plate features the µGasket negatives, the bottom plate, the microchannel structures and 1 mm pins (ejector pins, DLC, HASCO Hasenclever GmbH + Co KG, Lüdenscheid, Germany) to structure through holes for the ports. The tool was manufactured by CNC-milling (Biesinger GmbH, Haigerloch, Germany). To align both mold plates to each other, they were equipped with alignment pins.

The master mold embosses the µGaskets and the microfluidic channels into the SEBS/PMMA hybrid module while simultaneously sealing the interface between PMMA and SEBS. To prevent material accumulation and ease alignment, the ports were pre-cut in the 250 µm thick PMMA using a CO_2_ laser (VLS2.30, Universal Laser Systems, Scottsdale, AZ, USA) and the layer aligned on the steel mold with channel structures and the pins (Figure 2A,B; Table 1).

An SEBS sheet was laminated on the µGasket side of the steel mold using a handheld pressure roller (Steinel, Herzebrock-Clarholz, Germany).

For height control of the hot embossed chip layers, 600 µm spacers were placed between the two mold parts. The plates were stacked together and transferred into a hot press (Lab-Manual 300, Fontijne Presses, Delft, The Netherlands) preheated to 130 °C. The stack was pressed for 20 min while maintaining a pressure of 0.6 MPa and subsequently cooled down to 40 °C (Figure 2C; Table 1). The tool was removed from the hot press and opened to remove the embossed layer.

Another 750 µm SEBS sheet was flattened and laminated to a PET foil (3M^TM^ Scotchpak^TM^ 1022 Release Liner Fluoropolymer Coated Polyester Film) in a Compact Nanoimprint Tool (CNI v3.0, NIL Technology ApS, Kongens Lyngby, Denmark) using 2 bar at 100 °C for 7 min (Table 1). The SEBS sheet was rolled onto a 4-inch wafer using a handheld pressure roller followed by the PET foil. The SEBS/PET layer was peeled off from the wafer after lamination.

For the chip bottom layer, the flattened SEBS with PET back cover was laminated onto the PMMA channel side of the embossed SEBS/PMMA-hybrid layer using the handheld pressure roller (Figure 2D). The assembled connection chip was thermal fusion bonded in an oven at 67 °C overnight (Figure 2E). In the final step, the chip footprint was laser cut (Figure 2F).

### 2.2. Connection Strategy

Compression sealing leverages the leak-tight properties of SEBS, defining the connection mechanism of the µGaskets. The connection chip is compressed to the organ-chip module via a tailored spring-loaded fixation mechanism, thus increasing the surface contact to allow for van der Waals interactions of the mating chip surfaces. First, the connection chip was placed in a 3D-printed holder. A customizable organ chip with ISO-standard port locations was chosen for connection and aligned on top of the connection chip with the ports facing the µGaskets (ISO 22916:2022, International Organization for Standardization [26]). For fastening, a 3 mm PMMA bridge was placed on top of the connection fixed by two threaded rods in the frame of the 3D-printed holder. A spring (stainless steel compression spring 52/4/1, FK-Knörzer, Pfullingen, Germany) was loaded into each threaded rod on top of the PMMA bridge, followed by a knurled nut (M2 DIN466 stainless steel, Der Schraubenladen, Villingen-Schwenningen, Germany). Compression force was applied on the connection by a tightening of knurled nuts up to a pressure of 28.92 kN/mm^2^ on the two connecting µGaskets. This pressure was ensured by a spring deflection of 2.2 mm which was measured using a metric gauge block.

### 2.3. Fixation Force Measurement

The force of the fasteners on the µGaskets was measured using a thin piezoresistive force sensor (Tekscan A201 FlexiForce™ Piezoresistive Force Sensor, Norwood, MA, USA, 25 lb force specific). The sensor was placed between a 3D-printed holder and the connection chip and located underneath the µGasket structure. Clamping force on the connection was stepwise increased or decreased by the knurled nuts while recording the force on the sensor for each step.

### 2.4. Microstructure Analysis

For analysis of cross-sectional channel dimensions, side cuts were obtained using a cut-off machine (Struers S.A.S., Champigny-sur-Marne, France) and imaged using a stereomicroscope (Olympus SZX12 (Hamburg Germany) and ZEISS SteREO Discovery V12 (Oberkochen, Germany)). Mold channel structures were measured via a stylus profilometer (DektakXT, Bruker, Billerica, MA, USA). µGasket dimensions and surface roughness of both the hot embossed SEBS replica and the steel mold were measured using a 3D laser scanning microscope (Keyence VK-9710, Itasca, IL, USA).

### 2.5. Mechanical Testing of SEBS

Mechanical properties of the SEBS sheets were evaluated via tensile tests and shore hardness measurements. For the tensile test, dumbbell-shaped specimens with an overall length of 75 mm, gauge length of 20 mm, measurement width of 4 mm and thickness of 3 mm were molded from 750 µm SEBS sheets. According to ASTM D412 standard [27] for thermoplastic elastomers, the tensile test was performed using a universal testing machine (Z020, ZwickRoell GmbH & Co. KG, Ulm, Germany) at a speed of 500 ± 50 mm/min, until the specimen failed.

The shore hardness H_A_ was measured according to DIN ISO 48-4 using a durometer (HT-6510A, Schut Geometrical Metrology, Trossingen, Germany) at three different locations of a 6 mm specimen. The specimen was made by molding 750 µm SEBS sheets.

### 2.6. Burst Pressure Test

Bonding strength of the SEBS and PMMA layer was assessed in a burst pressure test by pressurizing with compressed air. Channel ports were sealed (UHU Plus Endfest, Baden, Germany) and luer connectors (BDMFTLL-9, Nordson MEDICAL) were attached to the inlet ports (using UHU Plus Endfest, Germany). After curing for a minimum of 24 h, the connection chip was attached to the gas line via the luer connectors and submerged in water to monitor gas leakages. A manual pressure controller was used to control a steady pressure increase of 0.5 bar every 10 s until chip failure was registered, or the maximum output pressure (8 bar) of the gas line was reached.

Cell culture conditions were resembled by pre-conditioning the connection chips at 37 °C and 95% humidity in an incubator (Heraeus BBD 6220, Thermo Scientific, Waltham, MA, USA) for two weeks followed by burst pressure tests. To account for a possible creep of the SEBS which may lead to deterioration of the sealing performance, connection chips were incubated in a connected state to two organ chips at cell culture conditions for 7 days followed by burst pressure tests.

The maximum internal working pressures sustainable by the µGaskets in the assembled state were determined through burst pressure testing, conducted with configurations comprising one and two additional interconnected chip modules, respectively. ISO-conform PMMA chips with a simple fluidic channel were selected as modules to be connected. To verify universality and compatibility of the µGasket sealing, additional organ chips with varying port sizes of 0.5 mm and 1.5 mm, as well as organ-chips made from PC and PET-G, were subjected to burst pressures tests. The connection was assembled by 2.2 mm compression of the springs which corresponds to a pressure of 29 N/mm^2^. The connection chip outlet ports were sealed and equipped with a luer connector at the inlet for the access of pressurized air. The chips were submerged in a water bath and the gas pressure increased by 0.5 bar every 10 s using a manual pressure controller until gas leakage was monitored.

Additionally, in the single chip connection, different fixation pressures were tested by closing or releasing the knurled nuts about three turns, correlating with additional 40 N or 26 N less clamping force.

### 2.7. Deformation Under Pressure

Fluorescence intensity of a dye perfused through the channel underneath the µGasket was used as a measure of the change in volume and thus the deformation of the channel, when the fixation was tightened. Therefore, a PMMA chip module was connected to the µGasket with a fixation pressure of 29 N/mm^2^ and perfused with 0.05 mg/mL FITC Dextran 4 kDa in PBS (Dulbecco’s Balanced Salt Solution w/o calcium w/o magnesium, Gibco, Thermo Fisher Scientific, Waltham, MA, USA) at 200 µL/h. The fixation was consecutively closed or opened by turning of the knurled nuts stepwise. For each turn, a fluorescence image of the channel underneath the µGasket was recorded using a fluorescence microscope (Observer 7, Carl Zeiss Microscopy, Oberkochen, Germany) and the fluorescence intensity was analyzed using Fiji.

For a thorough understanding of the compression behavior and its reversibility, two sequences of pressure change were tested: (a) tightening of the fasteners first, followed by loosening even below the starting point and back, and (b) loosening the fixation relative to the starting pressure, followed by tightening to the maximum spring compression.

### 2.8. Plug and Play Mechanism

The reusability of the µGasket connection for repeated module plugging and unplugging was tested with perfused colored water for 14 days. Two organ chips were fastened on the connection chip in the platform and detached and attached every second day and visually inspected for leakages. Effluent was collected for a perfusion of 200 µL/h to detect for minimal sealing failures. Two connection platforms with two detachable organ modules, respectively, were used for this experiment.

### 2.9. Image Analysis and Data Presentation Image Analysis

Dimensional measurements from microscopic images as well as fluorescence intensities were analyzed using Fiji (ImageJ version 1.54n). GraphPad Prism (Version 10.4.1, GraphPad Software, LLC., San Diego, CA, USA) was used for data presentation and statistical analysis. If not stated otherwise, data is given as mean ± standard deviation with sample sizes stated in each case.

## 3. Results

### 3.1. Facile and Robust Fabrication of Connection Chip with Integrated µGaskets

The µGaskets are designed such that they can be directly inbuilt into chips (Figure 1). Thereby, they enable a modular plug and play connection approach with minimized dead volume. The inbuilt µGaskets are SEBS-sealing rings, allowing a tight sealing of the chip interface without the need for connectors or other external materials. The chip consists of two main layers: (1) SEBS/PMMA-hybrid layer containing the µGaskets and microfluidic channel, (2) SEBS/PET bottom layer (Figure 1A,B).

The fabrication strategy is based on integrated, rapid structuring and bonding processes facilitating translation to industry-scale production (Figure 2). Laser cutting of the through holes into the PMMA revealed a highly scalable and precise technique. (Figure 2A). Notably, microstructure patterning and thermal fusion bonding are conducted simultaneously (Figure 2C). This reduces process time and costs and improves fabrication precision by avoiding erroneous alignment and post-processing of the layers. Pairs of long-lasting metal molds were used for hot embossing. Lamination of a smooth SEBS sheet for the final sealing of the chip allows a simple thermal fusion in a convection oven. The process is robust and does not require complex tools and processing steps.

Elasticity and durability of SEBS are major contributors to the sealing performance of the µGaskets. Mechanical characterization of the SEBS sheets used for the fabrication revealed an elongation at break ε_b_ of 703 ± 60% (*n* = 5) and a shore hardness A of H_A_ = 43.7 (*n* = 3). The measured values match the specification provided by the manufacturer of the SEBS pellets used for extrusion (ε_b_ = 700%, H_A_ = 40 [28]).

### 3.2. Integrated Hot Embossing/Thermal Fusion Process Results in Accurate Microstructures

The optimized two-sided hot embossing process achieved highly accurate features in both materials, namely the SEBS µGasket layer and the PMMA channel layer (Figure 3 and Figure 4). Side cuts of the hybrid modules demonstrated channel replica in PMMA with a height of 228 ± 11 µm (*n* = 12) and width of 442 ± 7.6 µm (*n* = 9). Master mold channel structures featured heights of 221 ± 2.1 µm (*n* = 2) and widths of 413 ± 5.7 µm (*n* = 2, Figure 3). The results confirm a structural transfer of approx. 96% and 93% accuracy.

Hot embossed SEBS µGaskets feature high embossing accuracy in all three dimensions and only minimal surface roughness (Figure 4). The µGaskets were designed in the original CAD drawing with a diameter of 3200 µm. Minor inaccuracies in the CNC-milling process result in mean diameter of the mold negatives of 3086 ± 20 µm (*n* = 4 gasket structures on the same mold). A high embossing accuracy of 99.9% was achieved for respective SEBS replicas with dimensions of 3089 ± 2.6 µm (*n* = 5, Figure 4B), as measured via light microscopy.

A reliable platform that allows a tight chip-to-chip connection is dependent on two main structural key factors. First, the appropriate height is needed to allow for compression of the sealing element to increase contact force and adapt to the micro-roughness of the counterpart. And second, a small surface roughness is needed to maximize contact area at the interface and enable van der Waals interaction. The hot embossed SEBS µGaskets feature heights of 388 ± 14 µm (*n* = 8, Figure 4B), as measured via laser scanning profilometry. Respective master molds measured step heights of 404 ± 13 µm (*n* = 8), resulting in an embossing accuracy of 96.0%.

With respect to roughness, the surface of the µGasket features an average roughness value *Ra* of 4.7 ± 0.75 µm (*n* = 8). While this low roughness already allows for reproducible and reliable chip-to-chip connection, further optimization of master mold (*Ra* of 4.8 ± 0.33 µm, *n* = 4) processing is expected to reduce surface roughness even further.

### 3.3. Robust Chip-to-Chip Connection Using a Spring-Loaded Fastening Mechnism

A spring-loaded fastening mechanism ensures controlled compression of the µGaskets. The tightness of the connection and the channel integrity were evaluated for a range of compression forces (Figure 5). Fluorescence intensity as a measure of channel volume was plotted against the applied fixation pressure at a flow rate of 200 µL/h, as used in experimental applications.

Linear compression was measured for the full range of the spring. The chip interface remained leak-free even at the lowest applied pressures. Fluorescence intensities remained >90% when increasing the clamping pressures stepwise from 29 N/mm^2^ to 83.4 N/mm^2^, confirming stable channels under the compressed µGaskets. Considerable intensity drops only appeared for compressions beyond the deflection capacity of the springs. However, the microchannels remained open and functional with fluorescence intensities of 42 and 53%. A pressure release to the starting condition restored the initial channel volume inferring the reversibility and reproducibility of the µGasket compression.

Even when reversing the order of pressure application, starting with compression release followed by compression increase, the sealing performance and the channel integrity both showed consistent behavior.

### 3.4. Tight Connection for a Wide Range of Internal Pressures

The strength of the overall system integrity of the connection chip was characterized by measuring the maximum burst pressure using pressurized air. Tests were conducted both with and without incubation at 37 °C and 95% humidity for 14 days, reproducing cell culture conditions used in OoC applications (Figure 5C). For both conditions, the chips were able to endure pressures of at least 8 bar, which is magnitudes higher than typical pressures during OoC experiments (in the range of a few hundred millibars [29]).

The strength of the sealing provided by the µGaskets was similarly characterized via burst tests (Figure 5C). The interface between the connection chip and the organ-chip module remained leak-free up to pressures of approx. 5 bar right after connection (“single” condition, *n* = 2) and up to approx. 3.5 bar after 14 days of incubation (“single-incubated” condition, *n* = 2). Two organ chips linked in series via the connection chip sustained pressures up to approx. 2.8 bar (“double” condition, *n* = 2) and up to approx. 3.6 bar after 14 days of incubation (“double-incubated” condition, *n* = 4). Incubation of the µGaskets in a configuration with two organ chips connected over 7 days did not deteriorate the sealing performance as confirmed by burst pressures of 3.1 bar (“double-incubated connection”, *n* = 5). Moreover, we tested the compatibility of the connection mechanism with different organ-chip port sizes (according to ISO standard) as well as with different chip materials of polycarbonate (PC) and polyethylene terephthalate glycol (PET-G): both 0.5 mm and 1.5 mm ports sustained pressures above 4 bar (4.2 bar (*n* = 4) and 4.4 bar (*n* = 4), respectively); connection of PC and PET-G chips remained leak-free up to 4.5 bar (*n* = 5) and 4.2 bar (*n* = 4), respectively. Furthermore, the compression force on the connection site using the spring-loaded fasteners was decreased or increased and tested for maximum burst pressures, respectively, using the same setup (Figure 5B). Connections with a lowered fixation pressure of 3 N/mm^2^ sustained internal channel pressures of 3 bar (*n* = 2), whereas higher clamping forces of 69 N/mm^2^ raised maximum resisted burst pressure to 6.5 bar (*n* = 2, Figure 5C). All burst pressures measured exceed typical OoC pressure ranges by a magnitude. The burst pressures were reached for a low measured fastening pressure of 29 kN/mm^2^, highlighting the excellent sealing capacity of the µGaskets.

### 3.5. The Plug and Play Mechanism Allows for Repeated Cycles of Disconnection and Tight Reconnection

Functionality of the reversible plug and play characteristic of the µGasket concept was confirmed by repeated detachment and reconnection of chip modules to the platform. Re-plugging of the organ chips via the µGaskets every second day did not show any leakage or wearing of the connection site (Figure 6). Tightness was shown for seven detachment cycles over 14 days under perfusion. Effluent collection verified completely tight chip-to-chip reconnections with a consistent flow in the system. Constant volumes of ~12.5 mL were collected after 48 h for platforms hosting two connected organ chips. Small variations in ±0.27 mL for MOC 1 and ±0.23 mL for MOC 2 are ascribable to errors of measurement of the weighted effluent.

## 4. Discussion

MOC systems provide a novel class of human-relevant in vitro models that open up entirely new avenues for studying inter-organ crosstalk, complex disease mechanisms and targets, as well as drug toxicities and efficacy [2,5]. Yet, interconnecting multiple organ modules can quickly become a highly complex process that is cost-, time- and labor-intensive. Hence, user-friendly and modular connection approaches are urgently required to ensure feasibility and flexibility in study designs. However, while modular approaches provide a lot of leeway in experimental design and timelines, they can be technically demanding in terms of dead volume, connector material, robustness or setup size [23,30]. Here, we overcome these challenges by developing a novel MOC connection strategy for the modular and flexible plug and play integration of the standardized organ module. Using on-chip integrated µGaskets, a robust and leak-proof interconnection with minimal dead volume is achieved.

The introduced connection strategy is based on a direct chip-to-chip interface tightened by integrated SEBS–µGaskets. The µGaskets enable leak-free sealing with seamless channel transition, thus avoiding the need for connectors in the system and minimizing the dead volume. SEBS is a medical-grade elastomer that benefits from a unique combination of thermoplastic and elastomeric properties with ideal sealing capability, low molecule absorption and good optical accessibility [31]. Depending on the blend of the polymer components as well as on the surface morphology, SEBS offers optical clarity with some blends up to transparency [30]. Furthermore, it exhibits chemical resistance to acids, tensides and alcohols such as methanol and ethanol [28].

An important aspect of the concept is the compliance with ISO standards with respect to dimensions and port locations, allowing the integration of a great diversity of standardized organ modules. Moreover, maintaining experimental flexibility not only allows flexible integration of arbitrary OoC modules but also respects tissue-specific timepoints in terms of individual connection and disconnection to the platform. The µGasket approach is specifically designed for a simple plug and play mechanism. Challenging the system’s tightness and durability upon repeated cycles of disconnection and reconnection of chip modules confirmed its eligibility for multi-organ studies where tissue-specific timepoints and analysis are needed. Minimal handling steps are required, and the sealing of a pre-wetted interface remains tight.

The user-friendly implementation of the connection strategy facilitates a wide range of applications. Thus, we developed a facile fabrication process with industrial-scale potential. Simultaneous two-sided hot embossing and thermal fusion bonding of both materials of SEBS and PMMA revealed precise microstructures while saving process time and alignment steps. µGaskets were embossed from a metal master mold with an accuracy of 96% for a height of 400 µm. Accurate gasket dimensions are important for reliable sealing performance. Likewise, a good surface roughness of the µGaskets is a fundamental requirement for a maximum surface contact area for tight compression sealing. Mean roughness values of 4.78 µm were achieved by hot embossing with a CNC-milled master mold. The leak-tightness of thus fabricated µGaskets was confirmed by a series of burst pressure and flow measurements. Although the SEBS’s inherent elasticity compensated for surface irregularities, improved surface quality of the master mold needs to be considered for future gasket generations. Surface post-treatments like polishing or powder coating are popular methods for improving surface roughness. However, material removal or coating is likely to alter feature dimensions on the mold [32]. By this means, a smooth surface finish must be weighed against accurate profile dimensions.

The hot embossed SEBS/PMMA hybrid layer combines the elastomeric sealing capacity with the necessary stability of microstructures upon compression sealing of the chip interconnection. Compression of the µGaskets to the organ chip is applied via spring-loaded fasteners. Measurement of the pressure in respect to the perfusable volume inside the microchannel located beneath confirmed a linear and controllable compression force while maintaining the channel integrity for a wide range of operating fixation conditions (0–137 N/mm^2^). This robustness of the system makes it functional regardless of user-specific implementation. Ease of connection and applicability of module interconnection are thus no longer limiting factors. In addition, the chip interface was leak-free for already low fixation pressures of only 3 N/mm^2^, highlighting the facile sealing efficiency of the µGaskets. This was further confirmed in a characterization of the connection strength by burst pressure tests. The µGaskets reveal tight chip-to-chip connections for different chip materials and port sizes for internal fluidic pressures of up to 5 bar, thus far exceeding typical OoC operating pressures of a few hundred millibar [29]. The achieved pressures are considerably higher than reported with other connection approaches such as the magnetic coupling of PDMS modules reaching hydraulic pressures of 0.3 bar [25]. Ultimately, challenging the system by both cell culture conditions and having additionally connected modules in the series did not compromise the interface tightness, making it a promising platform for multi-organ studies.

Further research and development of the concept could focus on a fastener click system with controlled fixation pressure to ensure a rapid, facile and reproducible connection. Refinements of the master mold for future µGasket generations could aim for smoother surfaces utilizing other manufacturing technologies such as selective laser melting or 3D-printing.

## 5. Conclusions

We developed a novel multi-organ connection strategy that overcomes current limitations of chip-to-chip interfaces: integrated µGaskets made of thermoplastic elastomers enable a tightly sealed fluidic interface of connected chips without the need for external connectors and without adding dead volume. The design and functionality of the µGaskets ensure interfacing of customizable organ chips with ISO-standard port locations. This customizability breaks down barriers for the combination of different organs and facilitates the use of already established organ models. The plug and play character of this connection mechanism adds flexibility in terms of the timepoint of connection and allows tissue-specific treatment in the multi-organ system. This approach enables model developers to focus first on replicating individual tissue models in a physiologically relevant manner and subsequent connection to other organ modules in a fit-for-purpose approach as it is required by the targeted question.

With respect to microfabrication, the integrated process featuring simultaneous microstructuring of the thermoplastic elastomer SEBS and thermoplastic PMMA as well as their thermal bonding saves processing time and is scalable to industry levels. The µGaskets sustained tight connections for a wide range of pressures while featuring a user-independent assembly without complex screwing but using spring-loaded fasteners.

In summary, we have developed a novel plug and play connection strategy with the potential to facilitate the use of MOC systems. The presented technology provides a highly flexible and user-friendly platform for the entire OoC community.

## Figures and Tables

**Figure 1 micromachines-16-01251-f001:**
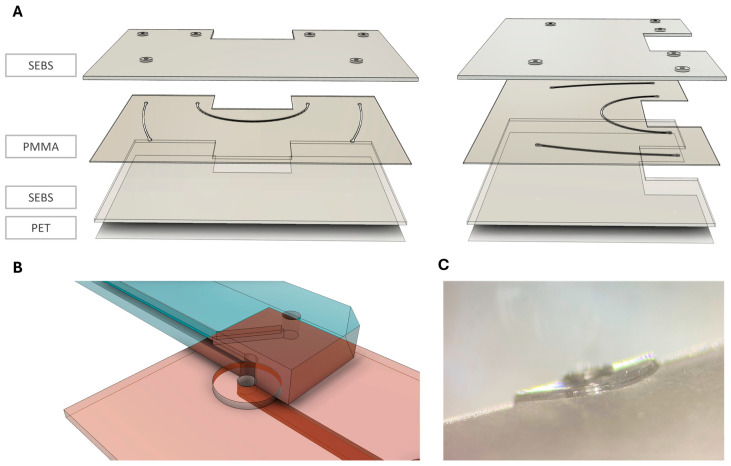
Exploded view of the multiple layers of the connection chip. (**A**) Layering of the connection chip; front view and side view. µGaskets are integrated in a 750 µm styrene–ethylene/butylene–styrene (SEBS) layer on top of a 250 µm polymethyl methacrylate (PMMA) layer with hot embossed microchannels. Chip bottom is made from 750 µm flattened SEBS together with a polyethylene terephthalate (PET) foil. (**B**) Sketch of the µGasket-to-organ chip interface with cutaway view of the organ chip. (**C**) Side view microscopy image of a µGasket.

**Figure 2 micromachines-16-01251-f002:**
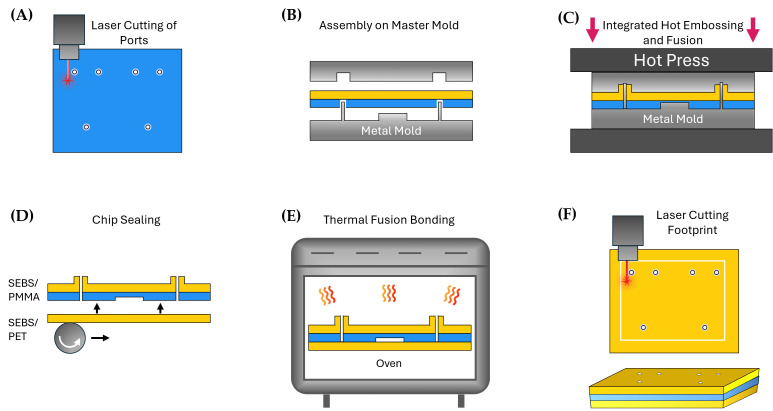
Fabrication process of a connection chip equipped with µGaskets. (**A**) Ports in the PMMA layer are laser cut before hot embossing. (**B**) Assembly of the laser cut PMMA layer and an SEBS layer in the metal master mold featuring the structures and pins for the microchannels and µGaskets. (**C**) Simultaneous microstructure patterning and thermal fusion bonding of the layers in a hot press. (**D**) Sealing of the hot embossed hybrid layer by lamination of another SEBS layer. (**E**) Thermal fusion bonding of the layers in an oven at 67 °C. (**F**) Laser cutting of the chip footprint.

**Figure 3 micromachines-16-01251-f003:**
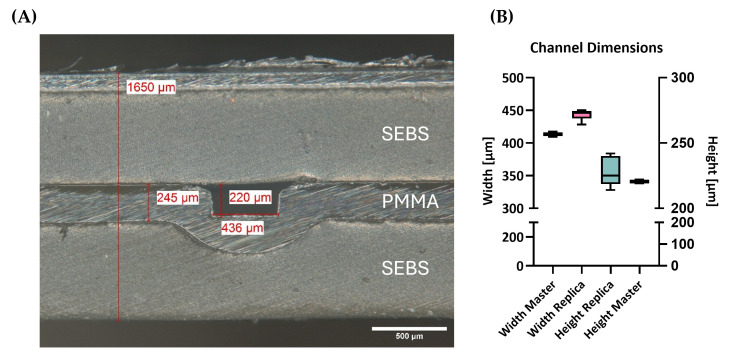
Microstructure analysis of the connection chip after fabrication by hot embossing: (**A**) microscopy image of the cross-sectional view of the microchannel; (**B**) measured channel dimensions of the hot embossed replica and the master mold, respectively.

**Figure 4 micromachines-16-01251-f004:**
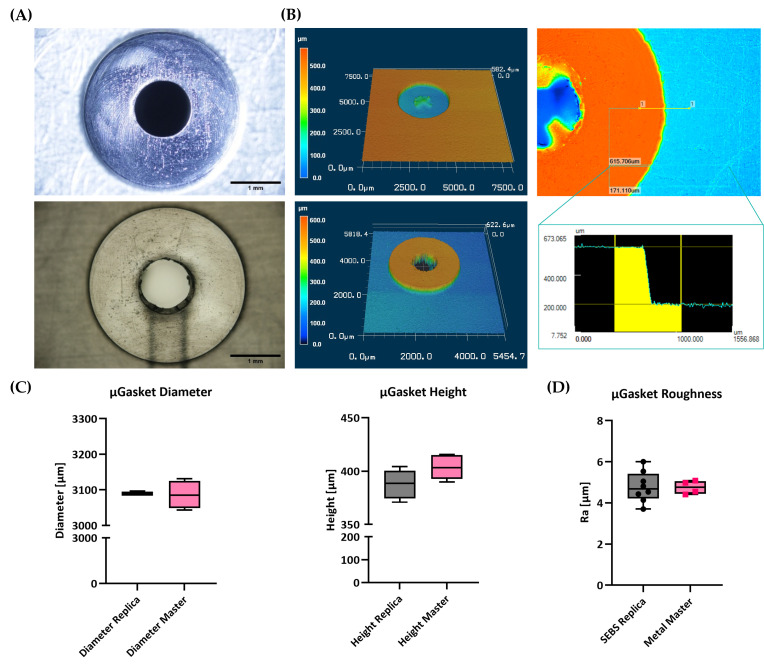
µGasket profiles and roughness. (**A**) Microscopic images of the µGasket master mold and the SEBS replica. (**B**) Laser scanning profilometry of µGaskets showing the profiles of master mold and SEBS replica and a step size measurement. (**C**) Comparison of the µGasket heights and diameters of the mold and replica. (**D**) Mean roughness of the µGasket surface on the master mold and the SEBS replica.

**Figure 5 micromachines-16-01251-f005:**
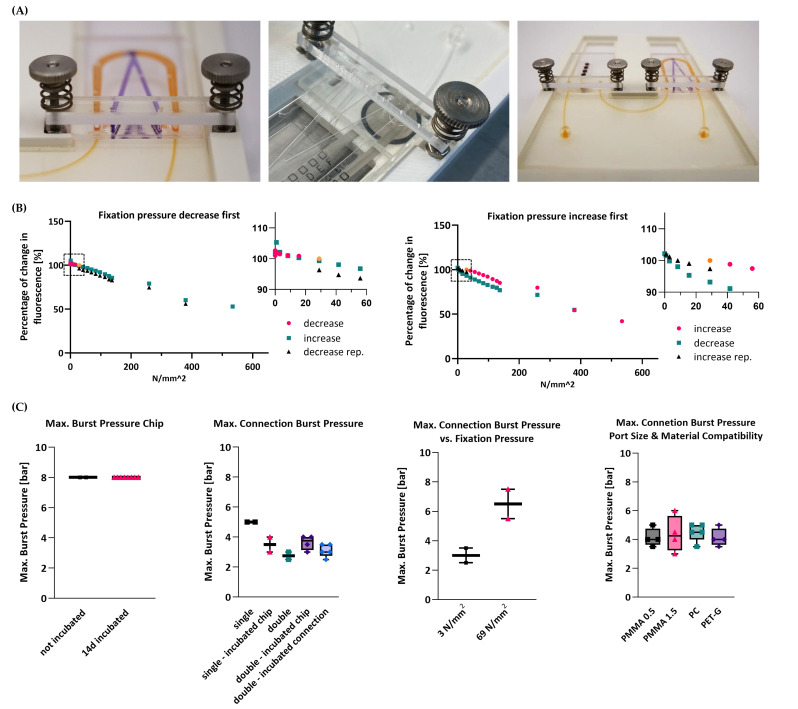
Analysis of connection fastening mechanism and maximum burst pressures. (**A**) Left: Image of the fastening mechanism of the connection of an organ chip to the connection chip. Middle: Image of the measurement of the pressure acting on the µGasket upon chip connection determined using a piezoresistive force sensor (Tekscan A201 FlexiForce™ Piezoresistive Force Sensor, 25 lb force specific). Right: Image of a connection chip connected to two organ chips in a 3D-printed holder. (**B**) Fluorescence intensity of the channel below the µGaskets for different fixation pressures as a measure of channel deformation. Shown is the sequence of pressure increase as first step followed by a release of the fixation clamps on the connection and vice versa as well as a magnified view of the data points for the operating pressures on the µGaskets (dotted frame), respectively. (**C**) Maximum burst pressures of the bonded connection chip with and without incubation in culture conditions for 14 days. Maximum burst pressures of µGasket connection to a single-organ chip and to two organ chips, with and without incubation of the connection chip and with the entire connected setup being perfused for 7 days. Maximum burst pressures of µGaskets connected to a single-organ chip with decreased and increased fixation pressure. Maximum burst pressures of µGaskets connected to single-organ chips with different port sizes as well as organ chips made of PC and PET-G.

**Figure 6 micromachines-16-01251-f006:**
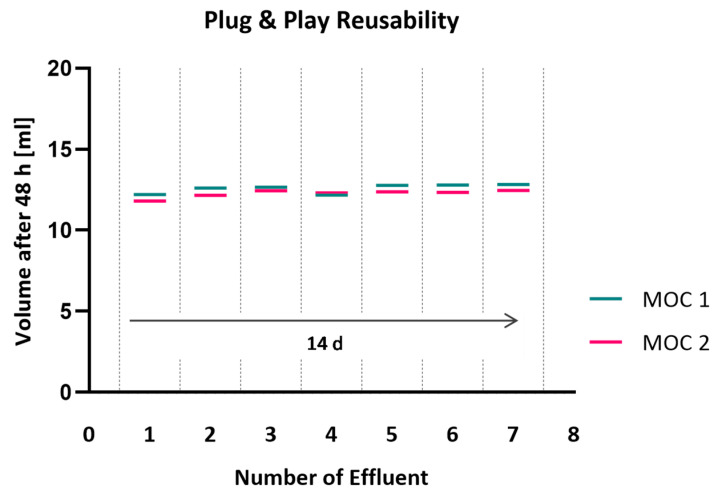
Perfused volume after repeated detachment and reconnection of organ chips. Weighing of collected effluent and visual inspection of perfused colored water was used to detect leakage.

**Table 1 micromachines-16-01251-t001:** Parameters of processes used for the fabrication of µGaskets.

Fabrication Steps	Microfabrication Technique	Tool	Parameters
Pre-cutting ports into PMMA	Laser cutting	CO_2_ laser (VLS2.30, Universal Laser Systems, Scotts- 134 dale, Scottsdale, AZ, USA)	Power: 20% PPI: 1000 Speed: 2%
SEBS/PMMA-hybrid µGasket layer	Hot embossing	Hot press (Lab-Manual 300, Fontijne Presses, Delft, The Netherlands)	Temp.: 130 °C Force: 0.6 MPa Holding: 20 min Cooling: to 40 °C
SEBS/PET bottom layer	Hot embossing	Compact Nanoimprint Tool (CNI v3.0, NIL Technology ApS, Kongens Lyngby, Denmark)	Temp.: 100 °C Force: 2 bar Holding: 7 min Cooling: to 40 °C
Bonding of µGasket layer and bottom layer	Thermal fusion	Oven (Memmert GmbH + Co. KG, Schwabach, Germany)	Temp.: 67 °C Time: ≥8 h

## Data Availability

The raw data supporting the conclusions of this article will be made available by the authors on request.

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
