# Peer review of "Plug and Play Multi-Organ Chips: Integrated µGaskets for the Facile and Reversible Connection of Individual Organ-on-Chip Modules"

_micromachines, 2025, doi:10.3390/mi16111251_

Round 1
Reviewer 1 Report
Comments and Suggestions for Authors
The authors have written an interesting piece on novel microfluidic elements that can be used to connect multiple organ-chips. They have characterized the new part thoroughly. In this reviewer’s opinion, this is a relatively small but valuable contribution to the field. This reviewer suggests some improvements be made to emphasize the significance of the authors’ contribution.
- Introduction: Elaborate on the relevance of multi-organ (or multi-tissue) models a bit more; and briefly mention their relative scaling.
- Introduction: Please expand the discussion of absorption of molecules into/onto materials a bit deeper, as this will strengthen the authors’ choice of material. Examples include 10.1039/b612140c, 10.1021/ac300771z, and 10.1016/j.bbrc.2016.11.062.
- Emphasize the novelty and significance of this work in the Introduction and Conclusion.
- Results: Include a functional study with two (or more?) organ or tissue models that were coupled using the µGasket.
- Include a photograph of the actual set-up, e.g. the one in the graphical abstract.
- Fig. 1 caption: Please give information on panels A through D.
- Materials and Methods, section Fabrication of µGaskets: Please provide all details on the procedures, including hot embossing programs (temperature ramps, hold steps, pressure applied – perhaps in a supporting file).
Reviewer 2 Report
Comments and Suggestions for Authors
The article proposed plug & play multi-organ-chips integrating μGaskets for the facile and reversible connection of individual organ-on-chip modules. The article can be considered for publication in "Micromachine" after revising the following questions. The comments are below.
- In the "materials and methods" section, the article mentioned that SEBS particles were used to make 750 μ m thick foil through external service provider extrusion, but the key performance parameters (such as shore hardness, elongation at break) of the SEBS foil were not clear. These parameters are crucial for understanding the sealing elasticity and durability of µ gaskets. It is recommended to supplement relevant data.
- The "burst pressure test" in the article shows that the connecting chip can withstand a pressure of at least 8 bar (much higher than the commonly used pressure of OOC), but the creep behavior of SEBS µ gaskets under high pressure is not discussed. In the cell culture environment (37 ° C, high humidity) for a long time, the creep of SEBS may lead to the deterioration of sealing performance. It is necessary to supplement the creep test data under different time lengths (such as 7 days, 14 days, 21 days) to verify its long-term stability.
- The article emphasizes that the µ gaskets connection strategy can adapt to "customized OOC module of ISO standard port", but does not provide specific compatibility verification data. For example, the connection tightness and fluid compatibility of commercial OOC modules with different materials (such as PDMS, PC) and different port sizes were not tested, which could not fully prove its "universality". It is recommended to supplement the docking experiment results of multiple types of OOC modules.
- In line 371 to 372, write variation of the weighted effluent.
- In line number 392 explains chemical inertness and optical accessibility of medical-grade elastomer.
- In line numbers 232 to 233, it would be better if colored water with different pH were tested.
- In line numbers 291 to 292, explain and give chemical reasons of van der Waals interaction.
- 8. In the introduction, the author can first introduce the progress of microfluidic technology and then focus on the research of Organ-on-Chips.
Chemical Engineering Journal, DOI: https://doi.org/10.1016/j.cej.2020.124700
Analytical Chemistry, DOI: https://doi.org/10.1021/acs.analchem.5c01644
Foods, DOI: https://doi.org/10.3390/foods14111928
Round 2
Reviewer 1 Report
Comments and Suggestions for Authors
The authors have made significant improvements to the paper, and addressed all of this reviewer's comments. This reviewer recommends publication in its current form.
Reviewer 2 Report
Comments and Suggestions for Authors
The author has completed the modification according to the comments pointed out by the reviewer.